# Highly Sensitive Temperature and Humidity Sensor Based on Carbon Nanotube-Assisted Mismatched Single-Mode Fiber Structure

**DOI:** 10.3390/mi10080521

**Published:** 2019-08-06

**Authors:** Weihao Yuan, Hao Qian, Yi Liu, Zhuo Wang, Changyuan Yu

**Affiliations:** 1Photonic Research Centre, Department of Electronic and Information Engineering, The Hong Kong Polytechnic University, Hong Kong SAR 999077, China; 2Photonic Research Centre, Department of Electrical Engineering, The Hong Kong Polytechnic University, Hong Kong SAR 999077, China

**Keywords:** miniaturized interferometer, carbon nanotube, fiber sensors, temperature, humidity

## Abstract

Here we report on a miniaturized optical interferometer in one fiber based on two mismatched nodes. The all-fiber structure shows stable performance of temperature and humidity sensing. For temperature sensing in large ranges, from 40 to 100 °C, the sensor has a sensitivity of 0.24 dB/°C, and the adjusted R-squared value of fitting result reaches 0.99461 which shows a reliable sensing result. With carbon nanotubes coating the surface of the fiber, the temperature sensitivity is enhanced from 0.24561 to 1.65282 dB/°C in a small region, and the performance of humidity sensing becomes more linear and applicable. The adjusted R-squared value of the linear fitting line for humidity sensing shows a dramatic increase from 0.71731 to 0.92278 after carbon nanotube coating, and the humidity sensitivity presents 0.02571 nm/%RH.

## 1. Introduction

The detection of environmental parameters, including physical and chemical changes, has become increasingly important for process control, security monitoring, and human protection. Temperature and humidity are the key parameters in these applications because of their great influence on food safety, manufacturing accuracy, and the service lives of precise instruments. Various methods have been used for temperature or humidity sensing, such as physical methods [1,2,3,4,5], chemical methods [6,7,8,9], and all-fiber devices [10,11]. Among these techniques, the all-fiber structures have great advantages, such as compact size, easy fabrication, strong electromagnetic resistance, relatively low price, and ability for distributed sensing [12,13]. On the strength of all-fiber configurations, diverse fiber sensors have been designed for various applications based on different mechanisms. Many fiber sensors have been reported, such as the Fiber Bragg Gratings sensor [14], long period fiber grating sensor [15], side polished fiber structure [16], hollow core fiber [17], single-mode–multimode–single-mode fiber structure [18], microcavity structure [19], Fabry–Perot fiber-optical sensor [20], up-tapered fiber [21], and mismatched fiber sensor [22,23]. Among these fiber sensors, the optical interference mechanism is widely used due to its high sensitivity and easy fabrication [12]. Combined with other materials or electrical devices, the fiber interferometer can be extensively applied in diverse areas.

Carbon nanotubes (CNT), which were first discovered by Iijima in 1991, are known as 1D nanostructures with sp^2^-hybridized carbon atoms [24]. The structure of CNT can be treated as a seamless cylinder which is constituted of one or several graphene layers (known as single-wall or multiwall CNT) [25]. CNT material provides unique advantages in many areas, such as high electrical and thermal conductivity, chemical stability, strong mechanical strength, and large surface-volume ratio [25,26]. Benefitting from its good thermal, electrical, and optical performance, CNT play a very important role in diverse fields, especially in sensing work. Many studies have been conducted for CNT-assisted sensors, such as strain sensors, gas sensors, humidity sensors, and temperature sensors [11,27,28,29,30,31]. In this paper, by combining the CNT material and a structured fiber device, we explore the sensing performance with temperature and humidity.

## 2. Experiments and Simulation Work

A mismatched structured fiber was successfully fabricated to serve as our miniaturized optical fiber interferometer (MOFI). The offset of the two single fibers was around 20 μm, and the distance between the two nodes was about 2 cm. The multi-walled CNT were bought from a commercial corporation. The CNT were dispersed in ethanol solvent and attached to the fiber structure using the drop-cast method. In order to make sure that the CNT adhered to the fiber firmly, an annealing process was conducted from the temperature of 100 °C to room temperature (24 °C).

To test the performance of the achieved mismatched fiber, and to explore the diverse potential applications based on light modulation by environment change, the experiments and comparison between the bare mismatched fiber and the CNT-assisted mismatched fiber were conducted on temperature and humidity sensing. The range of the temperature was controlled between 40 and 125 °C at an interval of 5 °C. The relative humidity (RH) sensing experiment was conducted by tuning RH between 45% and 95% at an interval of 10% at room temperature (24 °C). For each testing step of temperature and RH, extra observation time of 25 min was guaranteed to make sure that the results were stable and reliable. A wideband laser device (1300–1650 nm) was utilized as the light source, and the optical spectrum analyzer was used for spectra capture. Figure 1a is the schematic diagram of the mismatched single-mode fiber. Figure 1b,c displays the microscope image of the mismatched node and carbon nanotube coated part. 

The simulation work of the mismatched fiber structure was conducted, and the results are shown in Figure 2. BeamPROP was applied as the simulation tool, and the index profile type was set as STEP. The refractive index of core and cladding were 1.4504 and 1.4447, respectively, which are exact values of commercial single mode fiber. The offset value of two fibers was 20 μm, and the distance between the two mismatched nodes was 2 cm. The results show that the mismatched structure can successfully excite both core mode and cladding mode between the two mismatched nodes, which means that the MOFI has been achieved in one single fiber. As can be seen from Figure 2a–c, the core mode and multi-cladding modes were excited at the first node, and then the interference took effect at the second node. The constructive or destructive interference determined the optical intensity of the output light. Then the fiber structure with a thick cladding was simulated to compare with the normal cladding modes, which is shown in Figure 2d–f. It was found that the intensity of output light becomes very weak, which means that cladding modes dominate for this mismatched structure. With the change of external environment, the phase and intensity of the cladding mode can be tuned, which further gives rise to the change of output light intensity. Thus, this mismatched fiber structure can easily be used as temperature and humidity sensors because of the high sensitivity of cladding modes to the surrounding conditions. Due to the dominant cladding modes, the slight alteration of external parameters can be detected due to the optical modulation of cladding modes. The output signal will be influenced by the intensity change of cladding modes or the cladding’s phase alteration which will lead to more constructive or destructive interference.

## 3. Results and Discussion

Based on the MOFI structure, we can realize sensing performance through tuning light interference between the core mode and cladding modes which can be stably excited in the mismatched fiber structure. 

Figure 3 elaborates on the interference dip shifting with the continuous temperature change from 40 to 125 °C. The dip shift can be attributed to the length change of the fiber structure and the effective refractive index (RI) change of core and cladding areas which are caused by external thermal energy. The interference equation is shown below [32]:(1)$$I=I_{core}+\sum_{k}I_{cladding}^{k}+\sum_{k}2\cdot \sqrt{I_{core}\cdot I_{cladding}^{k}}cos\left[\frac{2\pi }{\lambda }\cdot \left(n_{eff}^{core}-n_{eff}^{cladding,k}\right)\cdot L\right]$$
I, $I_{core}$, $I_{cladding}^{k}$ mean the intensity of output light, core mode, and cladding modes. $n_{eff}^{core}$ and $n_{eff}^{cladding,k}$ mean the effective RI of core mode and cladding modes. *L* is the interference length.

The destructive interference condition is shown in the following equations [32]:(2)$$2\pi \cdot \left(n_{eff}^{core}-n_{eff}^{cladding,k}\right)\cdot L/\lambda =\left(2m+1\right)\pi$$
(3)$$\lambda =\frac{2\Delta n_{\mathrm{eff}}L}{2m+1}$$
$\Delta n_{\mathrm{eff}}$ means the difference of effective RI between core and cladding, and λ in Equation (3) means the wavelength of the interference dip. It is known that the effective RI of cladding has a relatively faster response to the temperature change compared to that of core, which means that the RI of cladding will increase firstly while the RI of core area will be unchanged at the start of temperature rising [33]. However, with time elapsing, the thermal energy around the fiber can finally give rise to the rising of core’s RI which can be larger than that of cladding’s RI [33]. This means that the RI difference between core and cladding will firstly decrease because of the RI rising on cladding and the unchanged RI on core. Moreover, it will then increase over a relatively long time in which the core’s RI can also be influenced by temperature change. Thus, except for the interference length change (*L* in Equation (3)), the wavelength shifting can also be attributed to alteration of the RI difference between core mode and cladding modes. The interference intensity will also be influenced by energy redistribution of core mode and cladding modes which is induced by RI change of fiber core and cladding. Based on the observation in the experiment process, the fiber condition can be stabilized within 90 s following the external environment change, which means that the response time of temperature sensing can be estimated to be 90 s. The sensitivity is calculated to be 0.2165 nm/°C which is almost four times higher than 0.0575 nm/°C reported by Dong et al. [34]. It was also found that in certain small temperature ranges, the interference intensity displays stable variation with the temperature change. This makes it possible for temperature sensing simply using a power meter without the need for an optical spectral analyzer. The wavelength of 1593 nm was chosen for measuring the relationship between the temperature and transmittance intensity. As can be seen from Figure 4, the relationship between temperature and interference intensity displays good linearity with the sensitivity of around 0.24 dB/°C. The adjusted R-squared value reaches 0.99461 which demonstrates the good linearity between temperature and output optical intensity.

To increase the sensitivity of temperature sensing performance, the CNT material was utilized to alter the cladding modes modulation of the MOFI structure. As can be seen from Figure 5a,b, without CNT coating, the intensity shows a small shift with the temperature altering from 40 to 34 °C. The alteration can be attributed to the deformation of the mismatched node that results in energy redistribution between core and cladding. With decreasing temperature, the mismatched node will shrink which can induce more energy to be coupled into the core area. Based on the simulation result in Figure 2, the cladding modes energy is dominate. This means that temperature change can lead to relatively equal intensity between core mode and cladding modes. The sensitivity increased dramatically when CNT material was attached to the fiber sensor. The enhancement can be due to the cladding mode modulation by CNT. The CNT can attach better with the fiber surface when temperature decreases which gives rise to a larger loss of cladding modes on the interface of fiber cladding and CNT. The linear fitting results between temperature and intensity of interference dip are displayed in Figure 6. The blue line shows the change of transmittance intensity with the alteration of temperature without CNT, and the red line indicates that CNT coating is present. It is obvious that the slope of the fitting line increases after CNT coating, which means that attaching CNT leads to the enhancement of temperature sensing sensitivity. The sensitivity is enhanced from 0.24561 dB/°C without CNT to 1.65282 dB/°C with CNT, and this value is almost four times higher than the 0.437 dB/°C that was reported by Yin et al. [35]. The adjusted R-squared value also shows slight augmentation from 0.92636 to 0.9738, demonstrating a stronger reliability of the sensing result. 

Based on cladding modes modulation, this mismatched fiber structure can also be used for humidity sensing, as shown in Figure 7. The experiment was conducted by changing RH from 45% to 95% at an interval of 10% at room temperature (24 °C). As we can see from Figure 7a, with the increase of RH, the interference dip shows a red shift with the range of 5 nm. This is due to the phase change of cladding modes induced by humidity alteration. The increasing humidity can lead to more sufficient contact between water molecules and the fiber surface which results in the effective RI change of cladding modes. As can be found from Equation (3), the change of $\Delta n_{\mathrm{eff}}$ will give rise to the interference dip shifting. Although the dip wavelength presents a relatively large shift of 5 nm following the humidity changes, the result is not linear enough to show the difference from 45% to 95%. In order to increase the linearity between the wavelength shifting and humidity change, the drop-cast method was applied to attach the CNT onto the fiber structure between the two mismatched nodes. Benefiting from the tubular structure of CNT, the 1D material can absorb water molecules which will induce the RI change of CNT. The result is displayed in Figure 7b. It shows that the relationship between humidity change and wavelength shifting becomes more linear. This can be explained that the RI change of CNT is smaller and more stable following the change of humidity which can further give rise to a more stable change of effective RI of cladding modes, and thus, the wavelength shifting displays good stability. Figure 8 gives the linear fitting results between the RH and wavelength shifting of interference dip. Without CNT attached (blue line), the adjusted R-squared value of fitting line is only 0.71731, which means that the wavelength shifting cannot accurately present humidity change. This value is dramatically enhanced to 0.92278 after CNT coating is applied, as depicted by the red line in Figure 8. The fit result reveals that the CNT-assisted fiber sensor can strengthen the accuracy rating of humidity sensing. The CNT-assisted humidity sensor presents the sensitivity of 0.02571 nm/%RH, which is slightly higher than 0.0235 nm/%RH that was reported by Ma et al. in the RH range 40–70% [36]. In addition, when considering the continuous alteration of RH from 40% to 90%, the linearity of humidity response in our work is relatively enhanced compared to that of two pseudo linear regions, which indicates the more applicable and practical sensing performance of our sensor [36].

Based on the above results, it is manifested that both the core mode and cladding modes can be excited in the mismatched fiber structure. This structure shows good and stable sensing performance of temperature and humidity which is due to the modulation of cladding modes by the external environment. Due to the dominant cladding modes, the all-fiber sensor shows good sensing reliability and sensitivity.

## 4. Conclusions

In conclusion, a mismatched fiber structure was successfully created and simulated that demonstrated its application for sensing temperature and humidity based on cladding modes modulation. The sensing experiments were conducted with the structured fiber, and it displayed good temperature and humidity sensing performance. With CNT attached to the fiber, the sensitivity of temperature detection in small ranges was enhanced from 0.24561 to 1.65282 dB/°C, and the humidity sensing became more reliable for real application. The combination of the manufactured MOFI structure and CNT achieved cost-effective, sensitive, and reliable sensing for both temperature and humidity. 

## Figures and Tables

**Figure 1 micromachines-10-00521-f001:**
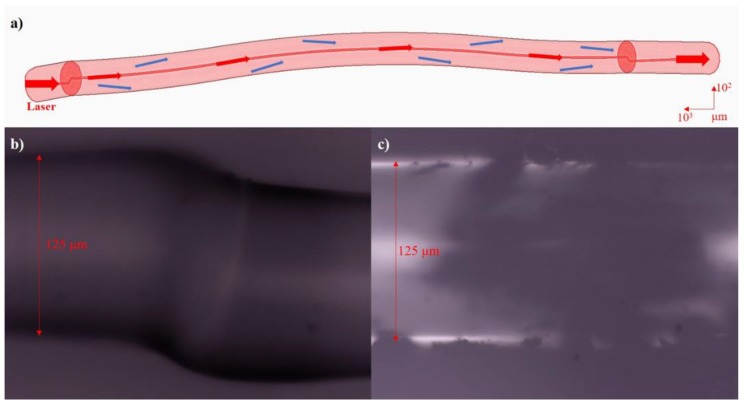
The schematic diagram of the mismatched single-mode fiber (**a**), the microscope image of the mismatched node (**b**) and the carbon nanotube (CNT) coated part (**c**).

**Figure 2 micromachines-10-00521-f002:**
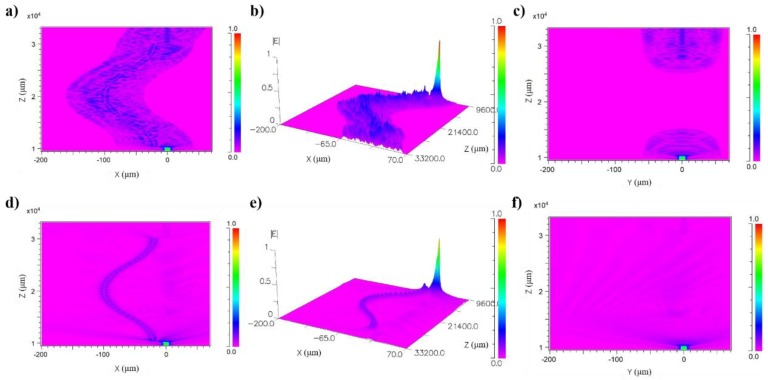
The simulation results of the mismatched fiber structure with normal cladding (**a**–**c**) and very thick cladding (**d**–**f**).

**Figure 3 micromachines-10-00521-f003:**
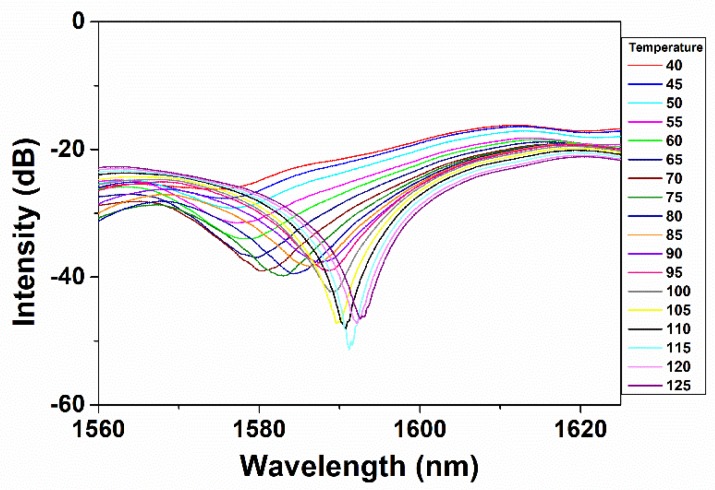
The relationship between interference dip wavelength shifting and the continuous temperature change from 40 to 125 °C at an interval of 5 °C.

**Figure 4 micromachines-10-00521-f004:**
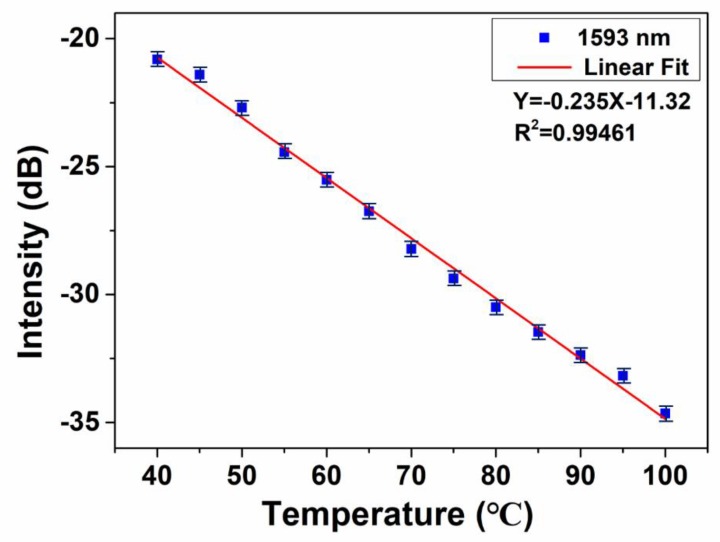
The relationship between temperature and interference intensity at the wavelength of 1593 nm.

**Figure 5 micromachines-10-00521-f005:**
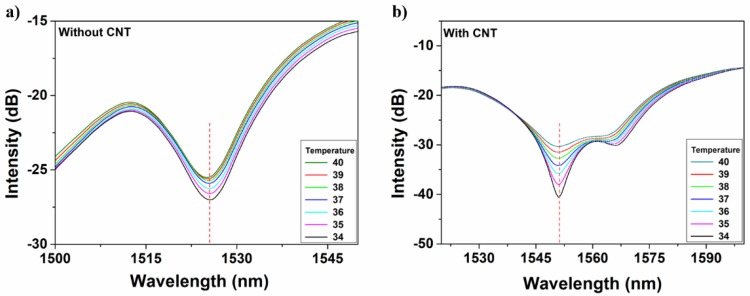
The spectra of temperature and interference intensity without (**a**) and with (**b**) CNT coating.

**Figure 6 micromachines-10-00521-f006:**
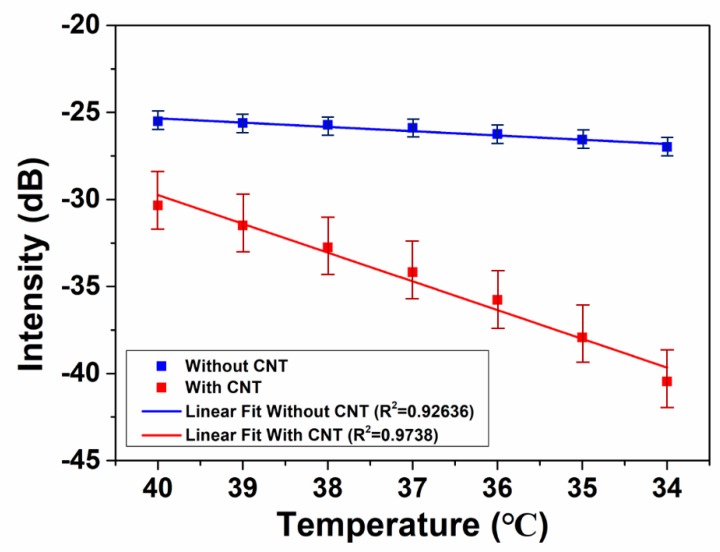
The relationship and linear fitting between temperature and interference dip intensity without (blue line) and with (red line) CNT coating.

**Figure 7 micromachines-10-00521-f007:**
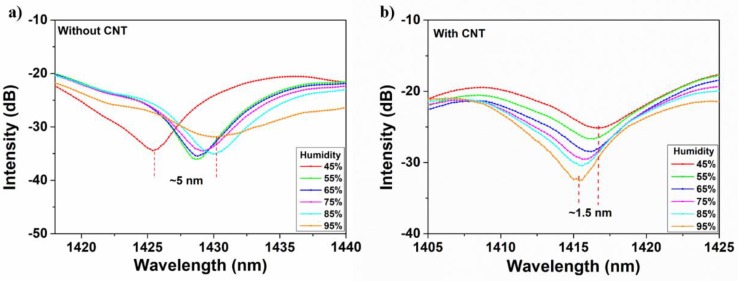
The spectra of relative humidity (RH) and interference wavelength without (**a**) and with (**b**) CNT attached.

**Figure 8 micromachines-10-00521-f008:**
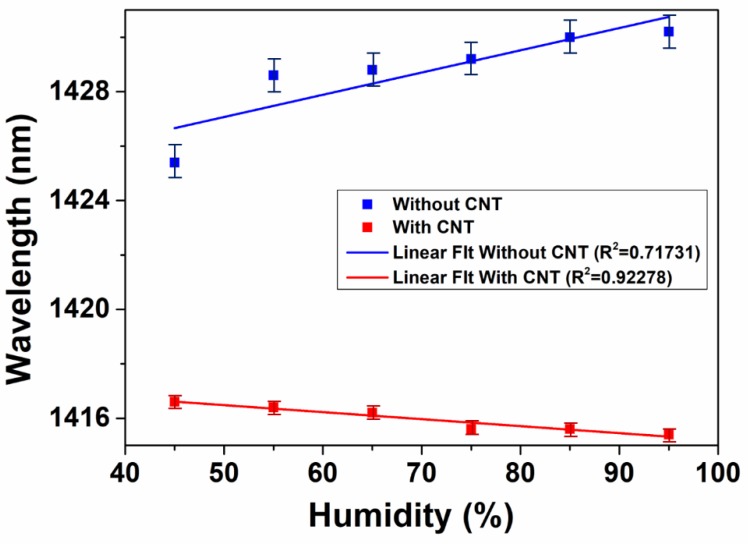
The relationship and linear fitting between RH and interference dip wavelength without (blue line) and with (red line) CNT attached.

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
