# Peer review of "Highly Sensitive Temperature and Humidity Sensor Based on Carbon Nanotube-Assisted Mismatched Single-Mode Fiber Structure"

_micromachines, 2019, doi:10.3390/mi10080521_

Round 1

Reviewer 1 Report

The manuscript is well written and shows relevant theoretical and experimental information on the use of CNT as a humidity and temperature sensor.

I suggest some contributions to the revised version:

- The introduction needs to be improved in order to show the reader the state of the art of moisture sensors, potential materials and key sensitivity results.

Some refs. are suggested below:

Wang, M.Y., Zhang, D.Z., Yang, A.J., Wang, D.R. and Zong, X.Q. (2019) Fabrication of Polypyrrole/Graphene Oxide Hybrid Nanocomposite for Ultrasensitive Humidity Sensing with Unprecedented Sensitivity. Journal of Materials Science: Materials in Electronics, 30, 4967-4976. https://doi.org/10.1007/s10854-019-00793-4 

Zhang, D., Chang, H., Li, P., Liu, R. and Xue, Q. (2016) Fabrication and Characterization of an Ultrasensitive Humidity Sensor Based on Metal Oxide/Graphene Hybrid Nanocomposite. Sensors and Actuators B: Chemical, 225, 233-240.https://doi.org/10.1016/j.snb.2015.11.024 

Araújo, E.S., Libardi, J., Faia, P.M. and de Oliveira, H.P. (2017) Humidity-Sensing Properties of Hierarchical TiO2:ZnO Composite Grown on Electrospun Fibers. Journal of Materials Science: Materials in Electronics, 28, 16575-16583.https://doi.org/10.1007/s10854-017-7571-5

Araújo, E.S. and Leão, V.N.S. (2019) TiO2/WO3 Heterogeneous Structures Prepared by Electrospinning and Sintering Steps: Characterization and Analysis of the Impedance Variation to Humidity. Journal of Advanced Ceramics, 8, 238-246.https://doi.org/10.1007/s40145-018-0309-x

- Some experimental configurations were only explained in the Results and Discussion section. Authors should explain all procedures, experimental techniques in the Experimental section.

I indicate a favorable opinion on the publication of the manuscript after the suggested revisions.

Reviewer 2 Report

In your article, the correlation between temperature and interference intensity is 0.24dB / C and R-squared value is 0.99461. But as everyone knows, the sensitivity of temperature to time is also very important. I wonder what your result is. I simply feel that there is a lack of engineering explanations such as "has fast response" or "increase quickly".

Reviewer 3 Report

The language in the manuscript can be improved significantly. This must be implemented throughout the manuscript.

See For example, on Page 1, "Carbon nanotube (CNT), which was firstly discovered by Iijima in 1991,..."  "The structure of CNT can be treated as a seamless cylinder....

The manuscript reads like a technical report.  The results and discussion component of the paper is weak; comparisons of the results with the literature is completely lacking....

In Figure 2, authors must present the details of the simulation; these include methods used, boundary conditions etc....

On page 2, the sentence should read as "In this paper, by combining  the CNT material and structured fiber device, we explore the sensing performance with temperature and humidity.

The sentence should read as "In order to make sure that the CNT is adhered to the fiber firmly, annealing process is conducted from the temperature of 100 ℃ to ?"

What are the error bars in the measurements in Figures 3-8?  How do these results compare with the literature?
